# Precision Therapy for Invasive Fungal Diseases

**DOI:** 10.3390/jof8010018

**Published:** 2021-12-27

**Authors:** Anne-Grete Märtson, Jan-Willem C. Alffenaar, Roger J. Brüggemann, William Hope

**Affiliations:** 1Antimicrobial Pharmacodynamics and Therapeutics, University of Liverpool, Liverpool L7 8TX, UK; william.hope@liverpool.ac.uk; 2Westmead Hospital, Sydney, NSW 2145, Australia; johannes.alffenaar@sydney.edu.au; 3Sydney Institute of Infectious Diseases, University of Sydney, Sydney, NSW 2145, Australia; 4Faculty of Medicine and Health, School of Pharmacy, University of Sydney, Sydney, NSW 2006, Australia; 5Department of Pharmacy and Radboud Institute for Health Sciences, Radboudumc and Radboudumc Center for Infectious Diseases, Radboud University Medical Center, 6525 GA Nijmegen, The Netherlands; roger.bruggemann@radboudumc.nl

**Keywords:** antifungals, pharmacokinetics, pharmacodynamics, precision therapy

## Abstract

Invasive fungal infections (IFI) are a common infection-related cause of death in immunocompromised patients. Approximately 10 million people are at risk of developing invasive aspergillosis annually. Detailed study of the pharmacokinetics (PK) and pharmacodynamics (PD) of antifungal drugs has resulted in a better understanding of optimal regimens for populations, drug exposure targets for therapeutic drug monitoring, and establishing in vitro susceptibility breakpoints. Importantly, however, each is an example of a “one size fits all strategy”, where complex systems are reduced to a singularity that ensures antifungal therapy is administered safely and effectively at the level of a population. Clearly, such a notion serves most patients adequately but is completely counter to the covenant at the centre of the clinician–patient relationship, where each patient should know whether they are well-positioned to maximally benefit from an antifungal drug. This review discusses the current therapy of fungal infections and areas of future research to maximise the effectiveness of antifungal therapy at an individual level.

## 1. Introduction 

Invasive fungal infections (IFI) are a common infection-related cause of death in immunocompromised patients. Even with the application of state-of-the-art diagnostic testing and deployment of modern antifungal therapy, the mortality of IFIs remains high [1]. Mortality in real-world settings is generally 20–50% but may be higher in specific contexts [2]. There are continuous threats, such as triazole resistance in *Aspergillus* spp., multidrug resistant *Candida auris*, and the emergence of new pathogens (e.g., *Emmonsia*, *Cryptococcus gattii*, *Exserohilum* etc.). Furthermore, invasive fungal diseases are increasingly seen in non-classical settings, such as critically ill patients, and as a complication of influenza, SARS-CoV-1 and SARS-CoV-2 [3]. 

The clinical pharmacokinetics of antifungal agents are generally well-characterised (Table 1). The pharmacodynamics for most antifungal agents against medically important *Candida* spp. and *Aspergillus* spp. are also reasonably well-characterised. This knowledge has resulted in a better understanding of desired regimens for populations, drug exposure targets for therapeutic drug monitoring (TDM), and establishing in vitro susceptibility breakpoints. Importantly, however, each is an example of a “one size fits all strategy”, where complex systems are reduced to a singularity that ensures antifungal therapy is administered safely and effectively at the level of a population. Clearly, such a notion serves most patients adequately but is completely counter to notions of precision therapy at an individual patient level. 

This viewpoint discusses the current therapy of fungal infections and areas of future research to maximise the effectiveness of antifungal therapy at an individual level. Reviews and guidelines of antifungal drug therapeutic drug monitoring are published elsewhere [11,15,21,22,23] and will not be considered in detail. 

## 2. Factors Affecting Precision Therapy

Clinical studies used for licensure involve a relatively homogeneous population—a cohort of patients that fulfil typically restrictive inclusion criteria. Patients with suboptimal outcomes that may benefit from a precision approach may only be apparent from post hoc analyses (e.g., patients with central nervous system disease) or from subsequent case reports and/or cohort studies obtained post-licensure. While such evidence is often biased, it helps identify clinical scenarios where a fixed dosing strategy may be inadequate and tailored, or an individualised approach is more appropriate. The following summarises the microbiological, pharmacological and clinical features where precision approaches may be considered.

### 2.1. Invading Pathogen

The appropriate therapeutic choices and the probability of therapeutic success are dependent on the causative fungal pathogen. The overall therapeutic response to antifungal therapy is generally comparable for different species of *Candida* and most *Aspergillus* species, but there are important exceptions. For example, a group of *Aspergillus* spp.—*A. lentulus*, *A. udagawae*, *A. viridinutans* and *A. fischeri*—are less susceptible to first-line anti-*Aspergillus* agents (e.g., amphotericin B, triazoles, echinocandins), thus limiting standard therapeutic choices [24]. 

Some fungal species display intrinsically reduced susceptibility to licensed antifungal agents. These include species less susceptible to triazoles (e.g., fluconazole versus *Candida parapsilosis*; fluconazole versus *Candida glabrata*), echinocandins (e.g., *C. parapsilosis*, *Trichosporon* spp., *Cryptococcus* spp., *Geotrichum* spp.) and polyenes (e.g., amphotericin B versus *A. terreus*) [25]. In these cases, accurate speciation enables informed choices about appropriate first-line agent(s).

In vitro susceptibility testing enables further refinement of therapeutic choices and decisions as to whether regimen intensification is a viable strategy. This should be performed by specialised laboratories using standardised techniques with results interpreted according to breakpoints published by the European Committee on Antimicrobial Susceptibility Testing (EUCAST) [26] or the Clinical and Laboratory Standards Institute (CLSI) [27] and potentially considering results from fungal genetic testing. Common resistance mechanisms resulting in reduced susceptibility to antifungal agents are summarised in Table 2. 

The effect of the relevant underlying resistance mechanisms (alone and in combination) is generally well-captured by the MIC. PK/PD targets for most of the triazoles against *Candida*, *Aspergillus*, and *Cryptococcus* are well-defined and can guide the likelihood of success with regimen intensification. A higher MIC requires a proportionally higher drug exposure to achieve a comparable therapeutic response. With regard to this, recently, EUCAST (European Committee of Antimicrobial Susceptibility Testing) has re-defined the intermittent susceptibility class to signal regimen intensification, which is required to secure a favourable clinical outcome [39]. The MIC and the associated PK/PD target provides an ability to objectively decide whether the patient is well-positioned to benefit from an antifungal agent.

### 2.2. Site of Infection

The success of therapy is dependent on the sites of infection. Examples of fungal diseases that are associated with poor clinical outcomes include infections within the central nervous system, endovascular infections, and disseminated disease [2]. Poor outcomes are a result of damage to vital structures and suboptimal partitioning of antifungal agents to the effect site. Bulky disease with significant tissue infarction and necrosis may further compromise drug penetration and make sterilisation with medical therapy alone impossible. Surgical resection may be required to debulk or completely remove acutely infected tissue and should be considered for large lesions that are contiguous and potentially compromise the heart, great vessels, and other mediastinal structures [40]. Suboptimal penetration of antifungal agents into fungal masses inside within pulmonary cavities pathognomonic of chronic pulmonary aspergillosis is associated with the emergence of antifungal resistance.

A high fungal burden may also be an important determinant of therapeutic response. Multifocal disease in a single organ is common (e.g., multiple *Aspergillus* nodules in the lung), and disseminated disease is probably significantly underdiagnosed. A high fungal burden may increase the probability of a resistant subpopulation being present at the time of treatment initiation and progressively expanding [41,42]. Unsurprisingly, a high fungal burden increases the time of sterilisation in the CSF and bloodstream in cryptococcal meningitis and taloromycosis, respectively [43,44]. 

### 2.3. Immunological Status of the Host 

There are many factors that are predisposed to invasive fungal diseases. Significant systemic immunocompromise is generally a prerequisite for development of disease (e.g., severe prolonged neutropenia, solid organ transplantation, HIV/AIDS) [45,46]. Increasingly, new biologics, such as inhibitors of tumour necrosis alfa (TNF-α) and interleukin (IL), are recognised as predisposed to histoplasmosis, aspergillosis, and invasive candidiasis [47]. Recently, the introduction of tyrosine kinase inhibitors (TKIs, e.g., ibrutinib indicated for B-cell cancers) has caused a rise of invasive fungal infections [48,49]. During ibrutinib therapy, atypical fungal infections have been reported—disseminated cryptococcosis, extrapulmonary *Pneumocystis jirovecii*, etc [48]. With the continuous emergence of new biological therapies, disseminated infections with atypical fungi are becoming more frequent, and treatment options are limited. Genetic polymorphism of the immune system has an important role in the development of invasive fungal infections. Polymorphism, deficiency and downregulation of specific genes (e.g., *PTX3, CX3CR1, STAT1, STAT3*) can suggest risk for fungal infection and potentially help stratify patients into high- and low-risk groups [50].

The main implication of the immunological deficit for the delivery of precision therapy is an objective assessment of the degree of underlying immunosuppression. Regimen intensification (e.g., drug dosing, PK/PD target attainment, combination therapy) may be required to affect a therapeutic response in a profoundly immunosuppressed host.

### 2.4. PK Variability 

The pharmacokinetics of antifungal drugs are typically highly variable because of absorption issues (e.g., food effects, differences between drug formulations), variation in protein binding (resulting in changes in free concentrations), drug–drug interactions and genetic polymorphisms in oxidative metabolism, errors in administration (e.g., crushing of posaconazole tablet [51]), and inflammatory status (in voriconazole therapy [52]). Importantly, however, a significant portion of observed variability remains unexplained [17,53,54,55,56,57,58]. The variability becomes clinically relevant for those agents with a narrow therapeutic index (voriconazole, itraconazole, posaconazole, amphotericin B and 5-flucytosine). In this context, the use of fixed regimens daily (including weight-based regimens) results in too many patients with concentration-dependent clinical failure and toxicity (e.g., voriconazole) [59]. This is the primary argument for routine therapeutic drug monitoring.

Finally, variability in drug exposure may result from poor compliance and a reduced amount of active substance within counterfeit drugs [60]. A thorough overview of the pharmacokinetics of antifungal drugs has been presented in multiple publications and is not discussed further here [17,58,61,62,63,64,65,66,67,68]. 

## 3. Delivering Precision Therapy

### 3.1. Scenario #1. No Cultures Available or Patient Is Culture Negative

This is perhaps the most common clinical scenario. Most patients in this group have significant underlying immunocompromise and possible invasive fungal infection (as defined by EORTC/MSG diagnostic criteria) [69]. Fungal cultures may have been obtained and be negative, or it may not have been possible to obtain deep cultures (e.g., because of thrombocytopenia). In this case, licensed antifungal therapy following international guidelines (e.g., ESCMID, IDSA, ECMM [14,16,70]) should be initially administered and regimens modified using therapeutic drug monitoring (TDM) as an adjunct where relevant. If appropriate, higher loading doses can be considered (e.g., CASPOLOAD study, micafungin in obesity) [71,72]. In addition, genotype-guided dosing has been shown beneficial in voriconazole therapy, where knowing the CYP genotype can help in deciding the initial dose [70,73]. 

In the absence of any microbiological data, there is no alternative but to use TDM drug exposure targets that have been defined for populations (Table 1). If there are specific clinical concerns (e.g., severe immunocompromised status) and regimen intensification is considered, it may be reasonable to use the top end of the range as the treatment target. In some cases, it may be reasonable to push beyond the upper bound of the target range if safety can be closely monitored and the anticipated toxicity is clinically acceptable. There is increased interest in using advanced computational methods to achieve desired therapeutic targets in a rapidly and optimally precise manner (e.g., model informed precision dosing) [74,75,76,77]. 

The assessment of the clinical response is notoriously difficult and requires experienced clinical judgement. Clinical signs and symptoms are generally nonspecific and affected (confounded) by multiple comorbidities. Nonspecific biomarkers (e.g., CRP, procalcitonin) may be useful as well as specific biomarkers, such as galactomannan [78]. Radiological resolution is typically slow and may paradoxically worsen, especially upon recovery from neutropenia [79]. Functional imaging with 2-fluorodeoxyglucose positron emission tomography integrated with computer tomography (FDG-PET/CT) may be helpful to define the optimum duration of therapy but it is not widely used due to costs and availability [80,81]. 

### 3.2. Scenario #2. Optimising the Antifungal Regimen Using the Organism and MIC

In this scenario, the invading pathogen has been cultured, and an MIC is available. This is commonly seen in candidemia, endemic fungal infections, and cryptococcal meningitis, but remains frustratingly low for mould infections, such as invasive aspergillosis, and infections caused by Mucorales [82,83]. For some fungal genera and species, the correlation between MIC and the clinical outcome is still poorly defined because of a significant impact of the underlying disease and immune status on the therapeutic response. 

Standard TDM based on PK–PD targets can be used (Table 1) but are now be embellished by the MIC. However, the PK/PD targets for echinocandins and polyenes are not routinely used to guide dosage adjustment in clinical settings, because of an absence of evidence if this has a quantifiable impact on clinical response and/or safety. Knowledge of the MIC can be used to optimise the use of fluconazole for *Candida albicans*, where an AUC/MIC of 100 is a well-cited and commonly accepted drug exposure target [10]. The MIC can be used to refine the dosing of voriconazole to treat a range of fungal pathogens. A C_min_/MIC target of 2–5 has been estimated from a large number of fungal infections treated with voriconazole [84] and is useful in deciding treatment options for strains with MICs just beyond the breakpoint. 

### 3.3. Scenario #3. Combining PK with Fungal-Specific Biomarkers

Medical mycology has multiple examples where the response to antifungal therapy can be quantified in real-time using a variety of fungal biomarkers. Galactomannan can be used to follow the course of invasive aspergillosis (especially in murine models). Beta-glucan has prognostic value for invasive aspergillosis and invasive candidiasis. Finally, quantitative fungal cultures of *Cryptococcus* and *Taloromyces* in CSF and blood can be used to follow the treatment response of cryptococcal meningitis and talaromycosis, respectively [44,85]. Clearly, such as strategy is limited to the subset of patients that are culture positive or have a positive biomarker.

The trajectory of a fungal-specific biomarker may be affected by both the organism and/or host-specific factors (Figure 1, example of biomarker response). The relationship between drug concentrations, fungal growth, and drug-induced antifungal effects can be quantified using PK–PD mathematical models [86,87]. Fungal-specific factors that are relevant include the fungal genera and species, fungal burden, and in vitro susceptibility (as discussed above). Host-specific factors include the nature and specific type of host immunological deficit and determinants of PK variability and drug handling. A successful therapeutic response requires both aspects to be satisfactorily addressed. The use of a biomarker also enables the treating clinician to escape the use of drug exposure targets derived from a population of patients (i.e., the average drug exposure target). Each patient has a specific therapeutic requirement that is based on their own specific circumstances and the properties of the invading pathogen. The balance of that relationship is revealed by the trajectory of the biomarker. A patient who is receiving a seemingly adequate antifungal regimen but has a climbing biomarker clearly needs regimen intensification, a new drug, or a combination of agents. In contrast, a patient whose biomarker settles quickly with treatment might feel more sanguine even if drug exposures are lower than considered ideal from estimates derived from populations.

An integrated approach addresses “true individualised therapy” or truly “precise therapy”. Each patient has their own individual drug exposure target, which is formed from a combination of host- and pathogen-specific factors. The use of both pharmacokinetic and pharmacodynamic data enables patients to understand whether they are well-positioned to maximally benefit from an antifungal drug and guide treatment decisions tailored for that individual.

## 4. Conclusions

With the increasing number of immunocompromised patients in the world, treatment of fungal infections will become progressively important. Here, we have presented the current approach to the diagnostics and optimisation of therapy and potential future directions. Whole genome sequencing, biomarkers, disease progression modelling, and FDG-PET/CT may all have an important role in the precision dosing of antifungal agents.

## Figures and Tables

**Figure 1 jof-08-00018-f001:**
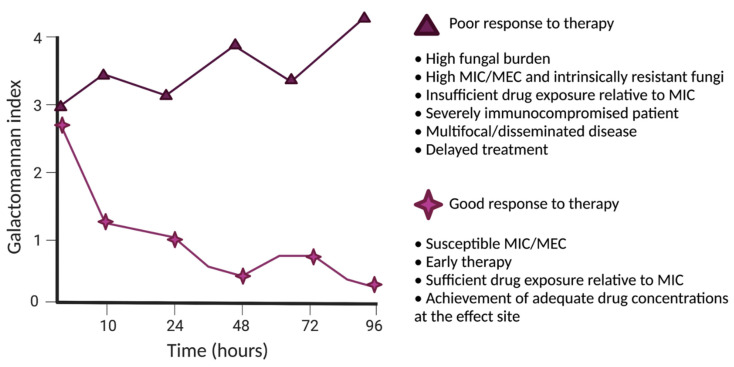
A schematic illustrating the use of a biomarker to guide antifungal therapy. The sold stars and triangles depict a favourable and suboptimal response to antifungal therapy, respectively. Figure created with Biorender.com, accessed on 13 December 2021.

**Table 1 jof-08-00018-t001:** TDM of antifungal drugs.

	PK and PK/PD Index Targets	References
Echinocandins		
Caspofungin	AUC/MIC = 450–1185 ^a^ Clinical target undefined	[4]
Micafungin	AUC/MIC >285 ^b^ >3000 ^c^	[5,6,7]
Anidulafungin	AUC/MIC = 123–2033 ^a^ Clinical target undefined	[8,9]
**Triazoles**		
Fluconazole	AUC/MIC > 100 ^d^	[10,11,12,13]
Itraconazole	C_min_ > 1 mg/L ^d^ C_min_ < 5 mg/L ^e^	[14,15]
Posaconazole	Prophylaxis: C_min_ > 0.7 mg/L ^d^ Treatment: C_min_ > 1–1.25 mg/L ^d^ AUC/MIC~ 200 ^f^	[14,16,17,18]
Voriconazole	C_min_ > 1 mg/L ^d^ C_min_ < 4–6 mg/L ^e^ C_min_/MIC = 2–5 ^g^	[11,14,19]
Isavuconazole	Clinical target undefined	[15]
**Amphotericin B**	Clinical target undefined Likely considerable differences between formulations	[15,19]
**5-flucytosine**	C_max_ < 100 mg/L ^e^	[15,20]

TDM, therapeutic drug monitoring; PK, pharmacokinetics; PD, pharmacodynamics; C_min_, trough concentration; AUC, area under the concentration-time curve; C_max_, maximal concentration; MIC, minimal inhibitory concentration; ^a^. *Candida glabrata*, *Candida albicans*, *Candida tropicalis, Candida parapsilosis* in murine models of disseminated candidiasis; ^b^. *Candida parapsilosis* (invasive candidiasis); ^c^. non-*Candida parapsilosis* population (invasive candidiasis); ^d^. Efficacy; ^e^. Toxicity; ^f^. *Aspergillus* spp.; ^g^. determined against invasive infections caused by medically important yeasts and moulds.

**Table 2 jof-08-00018-t002:** Antifungal drugs resistance mechanisms.

Drug Class	Drug Resistance	References
Triazoles	Inhibition, low binding affinity of enzyme 14α-demethylase through mutations (Erg11 p, drug target) Overexpression of enzyme 14α-demethylase (drug target) Efflux of antifungals through proteins in the major facilitator and ATP-binding cassette (CDR1-CDR5) superfamilies Genomic changes: loss of heterozygosity, segmental or chromosomal aneuploidies, chromosome copy number increase Combination of different mechanisms	[25,28,29,30,31]
Echinocandins	*Fks* mutations (substitutions of amino acids, Fks encodes the protein β-1, 3-glucan synthase) Not affected by transporters	[25,32,33,34]
Polyenes	Much less common than in azoles and echinocandins Mutations in genes involved in ergosterol biosynthesis (*ERG* genes) Proposed mechanism–decrease of polyene caused oxidative stress	[25,35]
5-flucytosine	*FUR1*, *FCY1*, *FCY2*, *UXS1* mutations	[36,37,38]

## Data Availability

Not applicable.

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
