# Peer review of "Precision Therapy for Invasive Fungal Diseases"

_jof, 2021, doi:10.3390/jof8010018_

Round 1
Reviewer 1 Report
This short paper focus on the concept of precision therapy applicated to invasive fungal diseases. Even if the paper is well written and developed a concept of high interest in the field of invasive fungal infections, there is several major concerns.
Major concerns
Line 36. Authors states ‘The clinical pharmacokinetics of antifungal agents are generally well characterised’, but it should be necessary that this statement is not really true for the most recent antifungals, ie isavuconazole and rezafungin. Their pharmacokinetics still need to be studied. The table 1 illustrated well this point since no therapeutic range was indicated for both recent drugs.
Line 73-75, Please precise which are the first-line anti-aspergillus agents
Table 1. Level of evidence of PK and PK/PD target indicated in this table should be discussed in the manuscript. Indeed, if therapeutic concentrations of several antifungal triazoles (voriconazole and posaconazole) are well defined by numerous works (including large prospective studies and meta-analysis), some PK and PK/PD targets proposed here are based on few studies conducted on small cohort for only some type of invasive fungal infections. For example, AUC proposed for echinocandins indicated only a value without range, which seems not very applicable in clinical pratice. Moreover, for some PK targets, it is indicated if the the threshold is useful for efficacy or toxicity, but such precision is not indicated for others PK or PK/PD targets. Why such differences ?
Legend of the table indicate some fungal pathogen for which therapeutic range or PK/PD target are proposed, but I think that this point need to be more detailed. For example, the number 1 confers to the PK/PD target AUC/MIC of 450, 865 and 1185 for Candida glabrata, Candida albicans, Candida parapsilosis, respectively. But the type of infection is not indicated. For voriconazole, a Cmin/MIC between 2 and 5 is proposed but one more time, type of infection and patients is not indicated (and reference seems to talk about combination of voriconazole and echinocandins).
In addition, the column ‘drug-drug interaction’ needs to be refine. It is not clear if this column indicate drug-drug interactions induced by antifungals (as indicated for example for azoles) or those that affect antifungal concentrations (as indicated for caspofungin) (or may be both ?). This column needs to be completed to precise this point.
Paragraph 3 (Immunological status of the Host). Authors should also focus on genetic susceptibility of invasive fungal infections (a subject recently reviewed in this paper : Naik et al, Front Genet 2021). Indeed, several genes encoded for proteins involved in innate or acquired immunity are prone to genetic variants that could affect the risk of fungal infections. Some of these genetic variants has been proposed to identify high-risk/low risk patients of invasive fungal infections and so initiate or not a prophylactic antifungal therapy (such evaluated in a current prospective interventional multicenter trial : NCT03828773).
Paragraph 4 (PK variability). Authors should add inflammatory status as an important factor of variability of voriconazole.
Line 142. Authors stated that ‘The variability becomes clinically relevant for those agents with a narrow therapeutic index.’ Among all antifungal drugs detailed in Table 1, which are those that have narrow therapeutic index ? For some antifungals, toxic concentrations are not well defined, so the term of narrow therapeutic index should not be used for all antifungals.
Line 147. Authors should also add, among the numerous factors of variability, the errors of administration (destruction of gastroresistant tablet for example for posaconazole : Mason MJ et al. Antimicrob Agents Chemother. 2019)
Figure 1. High MIC/MEC and intrinsically resistant fungi could be merged ?
In addition, please add delayed treatment among the factors involved in poor response.
In the second part entitled ‘Delivering Precision Therapy’, authors should cite and discuss recent works focusing on optimisation of initial doses of antifungal drugs (exemples : Voriconazole : Hicks et al, Clinical Pharmacology and Therapeutics ; Patel et al, Clinical Pharmacology and Therapeutics 2020 ; Caspofungin : Bailly et al, Antimicrobial Agents Chemotherapy 2020…etc)
Minor concerns
Table 1, abbreviations should be defined
Latine expression such as in vitro should be italic
Line 124, a point should be added after references 19 and 20
Line 141, please add genetic before polymorphisms and precise the involved genes
Line 148, please replace thorough by through
Author Response
Dear editor and reviewer,
Thank you for these comments and suggestions. We have addressed these accordingly below.
Best wishes,
Anne-Grete Märtson, on behalf of all the co-authors
This short paper focus on the concept of precision therapy applicated to invasive fungal diseases. Even if the paper is well written and developed a concept of high interest in the field of invasive fungal infections, there is several major concerns.
Major concerns
- Line 36. Authors states ‘The clinical pharmacokinetics of antifungal agents are generally well characterised’, but it should be necessary that this statement is not really true for the most recent antifungals, ie isavuconazole and rezafungin. Their pharmacokinetics still need to be studied. The table 1 illustrated well this point since no therapeutic range was indicated for both recent drugs.
Response
Thank you, indeed this is not entirely true, however isavuconazole has been quite extensively characterised. As also suggested by the second reviewer, we have decided to remove rezafungin as there are many more antifungals in the pipeline as described in the publication by Rauseo et al (Rauseo AM, Coler-Reilly A, Larson L, Spec A. Hope on the Horizon: Novel Fungal Treatments in Development. Open Forum Infect Dis 2020; 7: ofaa016. Available at: http://dx.doi.org/10.1093/ofid/ofaa016). As there are many more in the pipeline, these fall out of scope in our review.
- Line 73-75, Please precise which are the first-line anti-aspergillus agents
Response
We have added these to the sentence: For example a group of Aspergillus spp.—A. lentulus, A. udagawae, A. viridinutans and A. fischeri are less susceptible to first line anti-Aspergillus agents (e.g., amphotericin B, triazoles, echinocandins), thus limiting standard therapeutic choices [9].
- Table 1. Level of evidence of PK and PK/PD target indicated in this table should be discussed in the manuscript. Indeed, if therapeutic concentrations of several antifungal triazoles (voriconazole and posaconazole) are well defined by numerous works (including large prospective studies and meta-analysis), some PK and PK/PD targets proposed here are based on few studies conducted on small cohort for only some type of invasive fungal infections. For example, AUC proposed for echinocandins indicated only a value without range, which seems not very applicable in clinical practice. Moreover, for some PK targets, it is indicated if the threshold is useful for efficacy or toxicity, but such precision is not indicated for others PK or PK/PD targets. Why such differences?
Response
Thank you for this remark. Unfortunately, there is not enough information available for all the different antifungals and thus there is not sufficient data for all this information to be applied to routine clinical care. In addition, it is difficult to derive targets for the clinic as often doses are close to the top of the dose concentration relationship. We have edited the table accordingly and made specifications on the different indexes to make it clearer and more straightforward.
Legend of the table indicate some fungal pathogen for which therapeutic range or PK/PD target are proposed, but I think that this point need to be more detailed. For example, the number 1 confers to the PK/PD target AUC/MIC of 450, 865 and 1185 for Candida glabrata, Candida albicans, Candida parapsilosis, respectively. But the type of infection is not indicated. For voriconazole, a Cmin/MIC between 2 and 5 is proposed but one more time, type of infection and patients is not indicated (and reference seems to talk about combination of voriconazole and echinocandins).
Response
Thank you for this specification, we have added these specifics to the legend in Table 1. The Cmin/MIC ratio for voriconazole has been developed using data from different invasive fungal infections – e.g., invasive aspergillosis, non-neutropenic candidemia.
Table 1 legend:
1.Candida glabrata, Candida albicans, Candida parapsilosis respectively in a murine disseminated candidiasis model causing 1 log kill 24h; 2. Critically ill patients.; 3. Healthy adults; 4. Candida parapsilosis population (in invasive candidiasis or candidemia); 5.non-Candida parapsilosis population (in invasive candidiasis or candidemia); 6. efficacy; 7. Toxicity; 8. Different invasive fungal infections.
- In addition, the column ‘drug-drug interaction’ needs to be refined. It is not clear if this column indicate drug-drug interactions induced by antifungals (as indicated for example for azoles) or those that affect antifungal concentrations (as indicated for caspofungin) (or may be both?). This column needs to be completed to precise this point.
Response
Thank you, we discussed this, and decided to remove drug-drug interactions column from the manuscript, as al triazole cause a long list of interactions, act as substrates and inhibitors, thus it is difficult to capture this well in the review and the current column creates more questions than answers. We would like to refer to a thorough reviews for drug-drug interactions: Brüggemann et al in Lancet Hematology (https://pubmed.ncbi.nlm.nih.gov/34890539/) and Brüggeman et al CID (https://academic.oup.com/cid/article/48/10/1441/424268?login=true), which provide discussion and overview of these interactions that are out of scope for this viewpoint.
- Paragraph 3 (Immunological status of the Host). Authors should also focus on genetic susceptibility of invasive fungal infections (a subject recently reviewed in this paper: Naik et al, Front Genet 2021). Indeed, several genes encoded for proteins involved in innate or acquired immunity are prone to genetic variants that could affect the risk of fungal infections. Some of these genetic variants has been proposed to identify high-risk/low risk patients of invasive fungal infections and so initiate or not a prophylactic antifungal therapy (such evaluated in a current prospective interventional multicenter trial: NCT03828773).
Response
Thank you, this is important to mention in the manuscript.
Added to the section:
Genetic polymorphism of the immune system has an important role in the development of invasive fungal infections. Polymorphism, deficiency and downregulation of specific genes (e.g., PTX3, CX3CR1, STAT1, STAT3) can suggest risk for fungal infection and potentially help stratify patients into high and low risk groups 58.
- Paragraph 4 (PK variability). Authors should add inflammatory status as an important factor of variability of voriconazole.
Response
The variability of voriconazole PK is only partly explained by inflammation, however we have added this to the section:
The pharmacokinetics of antifungal drugs are typically highly variable because of absorption issues (e.g., food effects, differences between drug formulations), variation in protein binding (resulting in changes in free concentrations), drug-drug interactions and genetic polymorphisms in oxidative metabolism and inflammatory status (in voriconazole therapy 59).
- Line 142. Authors stated that ‘The variability becomes clinically relevant for those agents with a narrow therapeutic index.’ Among all antifungal drugs detailed in Table 1, which are those that have narrow therapeutic index? For some antifungals, toxic concentrations are not well defined, so the term of narrow therapeutic index should not be used for all antifungals.
Response
We have added a clarification to the text:
The variability becomes clinically relevant for those agents with a narrow therapeutic index (voriconazole, itraconazole, posaconazole, amphotericin B and 5-flucytosine).
- Line 147. Authors should also add, among the numerous factors of variability, the errors of administration (destruction of gastroresistant tablet for example for posaconazole: Mason MJ et al. Antimicrob Agents Chemother. 2019)
Response
The pharmacokinetics of antifungal drugs are typically highly variable because of absorption issues (e.g., food effects, differences between drug formulations), variation in protein binding (resulting in changes in free concentrations), drug-drug interactions and genetic polymorphisms in oxidative metabolism, errors in administration (e.g., crushing of posaconazole gastroresistant tablet [63]) and inflammatory status (in voriconazole therapy [64]).
- Figure 1. High MIC/MEC and intrinsically resistant fungi could be merged?
Response
We have changed the figure accordingly.
- In addition, please add delayed treatment among the factors involved in poor response.
Response
We have changed the figure accordingly.
- In the second part entitled ‘Delivering Precision Therapy’, authors should cite and discuss recent works focusing on optimisation of initial doses of antifungal drugs (exemples: Voriconazole :Hicks et al, Clinical Pharmacology and Therapeutics ; Patel et al, Clinical Pharmacology and Therapeutics 2020 ; Caspofungin : Bailly et al, Antimicrobial Agents Chemotherapy 2020…etc)
Response
Thank you for this suggestion, we have added to this section:
If appropriate, higher loading doses can be considered (e.g., CASPOLOAD study, micafungin in obesity)[85,86]. In addition, genotype-guided dosing has been shown beneficial in voriconazole therapy, where knowing the CYP genotype can help in deciding the initial dose [87,88].
Minor concerns
Table 1, abbreviations should be defined
Latine expression such as in vitro should be italic
Line 124, a point should be added after references 19 and 20
Line 141, please add genetic before polymorphisms and precise the involved genes
Line 148, please replace thorough by through
Response to minor concerns
Thank you for these suggestions, we have made the changes accordingly, however in line 148 thorough was correct word to use.
Reviewer 2 Report
1 - On Table 2, the antifungal class resistance mechanisms should include 5-flucytosine because this antifungal agent is an important agent for cryptococcal meningitis treatment. The monotherapy is not recommended due to the rapid development of resistance.
2 - If the authors want to include preclinical data for unapproved drugs, I agree with rezafungin, but it may be better to include promising candidates from other classes of antifungal as well. (i.e see recent review by Rauseo et al., 2020.) Rauseo AM, Coler-Reilly A, Larson L, Spec A. Hope on the Horizon: Novel Fungal Treatments in Development. Open Forum Infect Dis 2020; 7: ofaa016. Available at: http://dx.doi.org/10.1093/ofid/ofaa016.
Author Response
Dear reviewer,
Thank you for providing the comments, we have made changes accordingly.
Best wishes,
Anne-Grete Märtson, on behalf of all co-authors
- On Table 2, the antifungal class resistance mechanisms should include 5-flucytosine because this antifungal agent is an important agent for cryptococcal meningitis treatment. The monotherapy is not recommended due to the rapid development of resistance.
Response
Thank you, we have added 5-flucytosine resistance mechanism to Table 2.
- If the authors want to include preclinical data for unapproved drugs, I agree with rezafungin, but it may be better to include promising candidates from other classes of antifungal as well. (i.e see recent review by Rauseo et al., 2020.) Rauseo AM, Coler-Reilly A, Larson L, Spec A. Hope on the Horizon: Novel Fungal Treatments in Development. Open Forum Infect Dis 2020; 7: ofaa016. Available at: http://dx.doi.org/10.1093/ofid/ofaa016.
Response
Thank you for this suggestion, we have decided to remove rezafungin, as including all the novel antifungal drugs in development is out of scope for this manuscript. It is clear that not enough clinical data is available for most of these agents; thus we could not add additional information to the table.
Round 2
Reviewer 1 Report
I thank authors for their responses. Even if several of major concerns were solved, I think that various levels of evidence of PK and PK/PD targets indicated in the Table 1 needs to be refined (as already indicated in the first round of reviewing). Again, therapeutic concentrations (=PK targets) of several antifungal triazoles (voriconazole and posaconazole) are well defined by numerous works (including large prospective studies and meta-analysis), but not at all for others antifungals (echinocandin, isavuconazole especially) . For these latter, PK and PK/PD targets proposed here are based on few studies conducted on small cohort for only some type of invasive fungal infections (or even only in animal model for caspofungin ?). To solve this issue, I suggest to :
- indicate clearly in this table if proposed PK or PK/PD targets were only observed (as it seems to be the case for AUC of fluconazole and isavuconazole) or really related to effects (efficacy and/or toxicity). Authors indicated these precisions only for itraconazole, voriconazole and 5-FC. Why ?
- add a sentence of limitation of the strategy proposed in scenario 2 indicating that level of evidence of proposed PK/PD target is relatively low for most antifungals.
Additional minor comments :
- the legend of the table 1 indicates "for the azole... drug-drug interactions". This sentence should be moved since authors decided to remove drug-drug interactions of this table 1.
- what is the difference between Cmax and C2h for fluconazole ?
- date of the reference 35 is false
- reference 85 is not well indicated
Author Response
Dear editor/reviewer,
We have updated the Table 1, the responses are shown below.
Kind regards,
Anne-Grete Märtson, on behalf of co-authors
Comments:
I thank authors for their responses. Even if several of major concerns were solved, I think that various levels of evidence of PK and PK/PD targets indicated in the Table 1 needs to be refined (as already indicated in the first round of reviewing). Again, therapeutic concentrations (=PK targets) of several antifungal triazoles (voriconazole and posaconazole) are well defined by numerous works (including large prospective studies and meta-analysis), but not at all for others antifungals (echinocandin, isavuconazole especially) . For these latter, PK and PK/PD targets proposed here are based on few studies conducted on small cohort for only some type of invasive fungal infections (or even only in animal model for caspofungin ?). To solve this issue, I suggest to :
- indicate clearly in this table if proposed PK or PK/PD targets were only observed (as it seems to be the case for AUC of fluconazole and isavuconazole) or really related to effects (efficacy and/or toxicity). Authors indicated these precisions only for itraconazole, voriconazole and 5-FC. Why ?
- add a sentence of limitation of the strategy proposed in scenario 2 indicating that level of evidence of proposed PK/PD target is relatively low for most antifungals.
Response:
Thank you for these suggestions, we have edited Table 1. The healthy volunteer data has been removed as the exposure data can’t be used for TDM. In addition, we have updated the table to data that has been shown clinically and removed data that had lacking targets. The efficacy and toxicity are noted in the table, as well as clinical targets that are not specified are noted.
To scenario 2 we have added a sentence: However, the PK/PD targets for echinocandins and polyenes are not routinely used for guide dosage adjustment in clinical settings because of an absence of evidence this has a quantifiable impact on clinical response and/or safety.
Additional minor comments:
- the legend of the table 1 indicates "for the azole... drug-drug interactions". This sentence should be moved since authors decided to remove drug-drug interactions of this table 1.
Response:
This sentence has been removed.
- what is the difference between Cmax and C2h for fluconazole?
Response:
We have removed the Cmax targets due to uncertainty of these values.
- date of the reference 35 is false
Response:
This reference is now fixed.
- reference 85 is not well indicated
Response:
This reference is now fixed.